# Role of Impaired Mitochondrial Dynamics Processes in the Pathogenesis of Alzheimer’s Disease

**DOI:** 10.3390/ijms23136954

**Published:** 2022-06-23

**Authors:** Alexander V. Blagov, Andrey V. Grechko, Nikita G. Nikiforov, Evgeny E. Borisov, Nikolay K. Sadykhov, Alexander N. Orekhov

**Affiliations:** 1Laboratory of Angiopathology, Institute of General Pathology and Pathophysiology, Russian Academy of Medical Sciences, 8 Baltiiskaya Street, 125315 Moscow, Russia; nikiforov.mipt@googlemail.com (N.G.N.); drawnman@mail.ru (N.K.S.); 2Federal Research and Clinical Center of Intensive Care Medicine and Rehabilitology, 14-3 Solyanka Street, 109240 Moscow, Russia; avg-2007@yandex.ru; 3Petrovsky National Research Centre of Surgery, AP Avtsyn Institute of Human Morphology, 117418 Moscow, Russia; borisovevgenij5@gmail.com

**Keywords:** Alzheimer’s disease, neurons, mitochondrial dynamics

## Abstract

Mitochondrial dysfunction is now recognized as a contributing factor to neurodegenerative diseases, including Alzheimer’s disease (AD). Mitochondria are signaling organelles with a variety of functions ranging from energy production to the regulation of cellular metabolism, energy homeostasis, and response to stress. The successful functioning of these complex processes is critically dependent on the accuracy of mitochondrial dynamics, which includes the ability of mitochondria to change shape and position in the cell, which is necessary to maintain proper function and quality control, especially in polarized cells such as neurons. There has been much evidence to suggest that the disruption of mitochondrial dynamics may play a critical role in the pathogenesis of AD. This review highlights aspects of altered mitochondrial dynamics in AD that may contribute to the etiology of this debilitating condition. We also discuss therapeutic strategies to improve mitochondrial dynamics and function that may provide an alternative treatment approach.

## 1. Introduction

Alzheimer’s disease (AD) is a progressive neurological condition associated with neuronal degeneration, memory loss, learning disabilities, and significant changes in character and behavioral activity [1,2,3,4]. Disease progression increases with age and has been reported to affect 10% of people aged 65 to 75 and about 32% of people over 80 [5]. Alzheimer’s disease has also been reported in young adults, sometimes as young as 20 years of age (more likely to be caused by genetic mutations) [6]. By 2050, the number of people with AD worldwide is expected to reach 131 million, with the largest number of affected people expected in middle- and low-income countries [7].

AD slowly progresses into three identifiable clinical stages: mild (early), moderate, and severe (late), which exist alongside the normal aging process. AD is characterized by an insidious onset in a lucid state and memory problems. Short-term memory loss is the most typical initial symptom of Alzheimer’s disease. During AD, some changes gradually add to the initial symptoms, for example, changes in personality and behavior, deterioration in verbal communication, impaired visuospatial tasks, and motor dysfunction [6].

AD patients have common clinical and neuropathological features, including neuronal loss, intracellular neurofibrillary tangles (aggregates of hyperphosphorylated tau protein), and extracellular senile plaques composed of deposits of amyloid β (Aβ) that form as a result of proteolytic processing of the amyloid precursor protein. According to the totality of data, Aβ increases the vulnerability of neurons to oxidative stress and disturbances in the electron transport chain. Pathologically, AD is characterized by changes observed mainly in the neocortex, hippocampus, and other subcortical areas important for cognitive functions [8].

Over the past two decades, advances in pathogenesis have inspired researchers to explore new pharmacological therapeutics that are more focused on the pathophysiological phenomena of the disease. Currently available therapies, such as acetylcholinesterase inhibitors (rivastigmine, galantamine, donepezil) and *N*-methyl-d-aspartate receptor antagonists (memantine), have a minimal effect on the disease and target the late manifestations of the disease. These drugs slow the progression of the disease and provide symptomatic relief, but do not lead to a definitive cure [9]. While the neuropathological features of AD are known, the intricacies of the mechanisms have not been well defined. A lack of understanding of the pathogenic process may be a likely reason for the lack of effective treatment that can prevent the onset and progression of the disease. Thanks to significant progress in the field of pathophysiology over the past couple of years, new therapeutic targets have become available, which should contribute to the direct elimination of the underlying disease process [9]. Mitochondrial dysfunction and the disruption of mitochondrial dynamics have been an attractive target for therapeutic intervention in AD in recent years [10,11]. In this review, we will describe the current progress in understanding the mechanisms of altered mitochondrial dynamics in AD, as well as possible mitochondrial-targeted therapeutic strategies for AD.

## 2. Mitochondrial Functions. The Role of Mitochondrial Dysfunction in the Development of a Number of Chronic Diseases

Mitochondria are intracellular organelles containing a self-replicating genome [12]. Mitochondria perform key biochemical functions required for metabolic homeostasis and are the arbiters of cell death and survival. In eukaryotes, mitochondria generate energy in the form of ATP through oxidative nutrient metabolism, which can be divided into two main steps: (1) the oxidation of NADH or FADH2 produced during glycolysis, the tricarboxylic acid cycle, or fatty acid β-oxidation, and (2) direct oxidative phosphorylation to produce ATP. All these processes are regulated by a complex of transcription factors in mitochondria. Each mitochondrion contains from 800 to 1000 copies of mtDNA, which are maternally inherited and packaged into highly ordered nucleoprotein structures called nucleoids [13,14]. Although nucleoids are distributed throughout the mitochondrial matrix, they are often located near the cristae that carry the oxidative phosphorylation system. There is a small intermembrane space between the outer and inner membranes of mitochondria. The outer mitochondrial membrane and the intermembrane space are relatively more permeable than the inner membrane [15,16]. On the contrary, the inner membrane has a much more limited permeability, and contains enzymes involved in the processes of the electron transport chain and the formation of ATP [17,18]. The inner membrane surrounds the mitochondrial matrix, in which the electrons generated in the tricarboxylic acid (TCA) cycle are captured by the electron transport chain (ETC) to produce ATP. The electrochemical gradient generated on the inner membrane triggers the process of oxidative phosphorylation [19]. Most of the body’s cellular energy (>90%) is produced by mitochondria in the form of ATP through the tricarboxylic acid cycle and the electron transport chain [20].

The mitochondrial ETC is composed of five multisubunit enzyme complexes, I, II, III, IV and V, located in the inner mitochondrial membrane. Electrons donated by the coenzymes NADH and FADH2 to the TCA cycle are accepted and transferred to the ETC components in complex I (NADH-ubiquinone reductase) or complex II (succinate dehydrogenase), and then sequentially in complex III (ubiquinol-cytochrome c-reductase), complex IV (cytochrome c-oxidase) and, finally, oxygen through complex V (ATP synthase) [20,21]. This transfer of electrons along the electron transport chain combines with the transfer of protons across the inner membrane to create an electrochemical gradient that generates ATP. Mitochondria function continuously, metabolizing oxygen and generating reactive oxygen species (ROS). However, the flow of electrons through the ETC is an imperfect process in which 0.4 to 4% of the oxygen consumed by the mitochondria is not completely recovered, resulting in the production of ROS such as the superoxide anion, referred to as “primary” ROS [22]. The excessive formation of the superoxide anion additionally interacts with many other compounds and generates “secondary” ROS [23]. It has been previously established that ROS overproduction damages mitochondrial proteins and enzymes, membranes, and DNA. In addition, ROS can interfere with the ability of mitochondria to synthesize ATP and perform a wide range of metabolic functions, including fatty acid oxidation, the tricarboxylic acid cycle, the urea cycle, amino acid metabolism, and heme synthesis. Oxidative damage can increase the tendency of mitochondria to release cytochrome through the pores, leading to the activation of the apoptosis cascade. Potential pathological outcomes of ROS production also include the formation of mtDNA mutations or deletions, oxidative damage to the respiratory chain, lipid peroxidation, and general mitochondrial dysfunction [24]. Under normal conditions, ROS overproduction is limited in mitochondria to protect cell organelles from oxidative damage through enzymatic and non-enzymatic defense systems. On the other hand, when antioxidant defenses are compromised, there is an overproduction of ROS, which then leads to oxidative damage to proteins, DNA, and lipids in the mitochondria. This disrupts enzyme functions in the respiratory chain and ultimately leads to mitochondrial dysfunction, reduced mitochondrial biogenesis, and a wide range of neurodegenerative disorders. Apoptosis and excitotoxicity are two important causes of neuron death, and the role of mitochondria is crucial in both cases [20,22,25]. Increased ROS production during the neurodegenerative process can affect mitochondrial parameters, as well as ATP production, membrane potential, the activation of the permeability transition pore, and calcium uptake. These changes can lead to neuronal damage. The first evidence of the involvement of mitochondria in the pathogenesis of the neurodegenerative process was obtained by detecting a deficiency of complex I in the substantia nigra and platelet mitochondria in patients with Parkinson’s disease (PD). Additional strong evidence of the deficiency of the electron transport chain was found: complex I and cytochrome oxidase (complex IV) in AD and complexes II and III in Huntington’s disease (HD) [26,27,28].

## 3. The Importance of Mitochondrial Dynamic Processes for Maintaining Cell Vitality

The main dynamic activities of mitochondria are fusion (the connection of two organelles into one), fission (the division of one organelle into two), transport (directed movement within the cell), and mitophagy (directed destruction along the autophagic pathway). Dynamic behavior has been shown to play an important role in both normal physiology and disease states [29,30].

### 3.1. Fusion and Fission of Mitochondria

Mitochondrial fusion is an evolutionarily conserved process mediated in mammals by three large GTPases of the dynamin superfamily: mitofusinome 1 and 2 (MFN1 and MFN2), which are integral outer membrane proteins, and OPA1 proteins, which have multiple isoforms associated with the inner membrane [29]. Because mitochondria have double membranes, mitochondrial fusion is a two-step process requiring outer membrane fusion followed by inner membrane fusion. The fusion process allows the preservation of mitochondrial functions and maintains genetic and biochemical homogeneity, allowing a reduction in the concentration of superoxide and mutated mitochondrial DNA [31].

In addition to fusion, mitochondrial fission is also important for cellular and organismal physiology [29]. Mitochondrial division is mediated by DRP1 protein, a large GTPase that is recruited to the outer mitochondrial membrane through a set of receptor proteins (MFF, FIS1, MID49, and MID50) [32]. Once in the mitochondria, DRP1 gathers around the tubule and constricts it in a GTP-dependent manner, mediating cleavage [33]. In addition to influencing mitochondrial morphology, fission is involved in many functions, including the facilitation of mitochondrial transport, mitophagy, and apoptosis [34].

### 3.2. Mitochondrial Biogenesis

The generation of new mitochondria, mitochondrial biogenesis, differs from mitochondrial fission in that it involves the complete replication of mitochondrial DNA. Mitochondrial biogenesis is driven by the transcriptional activators NRF-1, NRF-2, and PGC-1α, which are activated in various pathways, such as receptor tyrosine kinases, natriuretic peptide receptors, and nitric oxide, through the formation of cyclic guanosine monophosphate (cGMP) [35]. The expression of nuclear-encoded mitochondrial genes is actively transcribed, and proteins with mitochondrial target sequences, including TCA and oxidative phosphorylation enzymes, antioxidant enzymes, and mitochondrial transcription factor A (TFAM), are imported into mitochondria via the translocase by the TOM/T I M protein complex [36].

### 3.3. Mitochondrial Transport

Mitochondria are transported within neurons from (anterograde transport) and to (retrograde transport) the neuronal soma by a mechanism known as axonal transport [37]. The mobility of mitochondria in neurons is necessary for the delivery of ATP to the sites of synapses, to stimulate the growth of axons, to buffer calcium, and to ensure the restoration and degradation of mitochondria [38]. Mitochondrial transport in neurons can be facilitated along microtubule pathways or actin filaments based on the cellular compartment. The structure and polarity of microtubules within axons and dendrites are different, with approximately 90% of microtubules oriented with their positive end away from the soma to the axon. In dendrites, microtubules have a mixed orientation and density at the proximal end of the soma, with polarity and organization becoming more axon-like at the distal sites [39]. To facilitate axonal transport, adapter proteins such as syntabulin, MIRO, and Milton are associated with motor proteins of the kinesin-1 and kinesin-3 families to transport mitochondria to the (+) end of microtubules in an anterograde direction. Protein complexes consisting of dynein and dynactin proteins guide mitochondria to the (−) end of microtubules, facilitating retrograde transport [40]. Thus, kinesins typically transport mitochondria in an anterograde direction in axons, while both kinesin and dynein can carry out the bidirectional movement of mitochondria in dendrites. Mitochondria can also move along actin filaments in dendritic spines, growth cones, and synaptic buds for short-term redistribution using myosin [41]. Although MIRO and Milton have been identified as mammalian adapters responsible for mitochondrial transport via kinesin, additional motor and adapter proteins are involved in axonal transport mechanisms, ensuring the proper distribution of mitochondria in the cell [42]. The general scheme of mitochondrial transport in neurons is shown in Figure 1.

### 3.4. Mitophagy

To maintain functional mitochondria, cells use autophagy, in which damaged organelles are removed through lysosomal degradation. For mitochondria-specific autophagic removal, known as mitophagy, the collapse of the mitochondrial membrane, or the release of mitochondrial-associated lipids, serves as a signal to stimulate the targeting of mitochondria to autophagosomes [43]. These autophagosomes then fuse with lysosomes for the major degradation and reuse of components that primarily occur in the cell body, although local autophagic processes in axons have also been identified [44]. The most common cause of autophagy is nutrient imbalance. A key sensor of energy homeostasis, AMP-activated protein kinase (AMPK), can activate or increase levels of mitophagy through the direct phosphorylation of unc-51-like activating kinase 1 (ULK1) during energy stress [45]. The best described mitochondrial quality control mechanism involves the PINK1/PARKIN pathway. The loss of mitochondrial membrane potential typically stabilizes the PINK1 protein on the outer membrane, which then phosphorylates proteins including MIRO, MFN, and ubiquitin [43]. This leads to the movement of the E3 PARKIN ligase, which leads to the activation of the autophagic mechanism. In neurons, most of the mitophagic degradation requires the transport of dysfunctional mitochondria within autophagic vacuoles from distal to sematodendritic regions. For this, autophagosomes fuse with late endosomes and form amphysomes together with dynein–snapin motors to transport autophagic vacuoles back into the soma [46]. Impaired retrograde transport affects the ability to remove dysfunctional mitochondria.

## 4. General Pathogenesis of Alzheimer’s Disease

AD is a complex and progressive neurodegenerative disease. Reported histopathological features of AD are extracellular aggregations of amyloid β (Aβ) plaques and intracellular aggregations of neurofibrillary tangles (NFTs) composed of hyperphosphorylated τau protein. The hyperphosphorylation of τau protein leads to microtubule destruction. Aβ plaques initially develop in the basal, temporal, and orbitofrontal areas of the neocortex of the brain, and in later stages spread throughout the neocortex, hippocampus, amygdala, diencephalon, and basal ganglia. In critical cases, Aβ is found throughout the midbrain, lower brainstem, and cerebellar cortex. This concentration of Aβ induces the formation of τ-tangles, which are found in the locus coeruleus, as well as in the transentorhinal and entorhinal areas of the brain. At a critical stage, it spreads to the hippocampus and neocortex. Aβ and NFT are considered to be major players in disease progression [47].

### 4.1. The Role of Beta-Amyloid in the Pathogenesis of Alzheimer’s Disease

The initial stages of AD development are associated with altered cleavage of the amyloid precursor protein (APP), integral to the plasma membrane protein, by β-secretases and γ-secretases to form insoluble Aβ fibrils. Aβ then oligomerizes, diffuses into synaptic clefts, and interferes with synaptic signaling. Hence, it polymerizes into insoluble amyloid fibrils, which coalesce into plaques. This polymerization leads to the activation of kinases, which leads to the hyperphosphorylation of the microtubule-associated τ-protein and its polymerization into insoluble NPCs. The aggregation of plaques and tangles is followed by the recruitment of microglia surrounding the plaques. This contributes to the activation of microglia and the local inflammatory response, and also leads to neurotoxicity [48].

### 4.2. Hyperphosphorylation of τau Protein in Alzheimer’s Disease

AD is also characterized by the presence of NFT. These tangles are the result of the hyperphosphorylation of the microtubule-associated τau protein [49]. NFTs are fragments of paired and helically twisted protein filaments in the cell cytoplasm of neurons, as well as in their processes. The τau protein has a microtubule-binding domain that interacts with tubulin to form mature and stable microtubules [50]. When the τ-protein comes into contact with released kinases, due to the abundance of Aβ in the environment, it is hyperphosphorylated. Its hyperphosphorylation leads to its oligomerization. The tubules become unstable due to the dissociation of their subunits, which disintegrate and then turn into large pieces of τ-filaments, which are further combined into NFT. These NFTs are straight, fibrillar, and highly insoluble regions in the cytoplasm and processes of neurons, leading to an abnormal loss of communication between neurons and signal processing and, finally, to apoptosis in neurons [48]. In addition, τau phosphorylation is regulated by several kinases, including glycogen synthase kinase 3 (GSK3β) and cyclin-dependent kinase 5 (CDK5), activated by extracellular Aβ. Although GSK3β and CDK5 are primarily responsible for the hyperphosphorylation of τau, other kinases such as protein kinase C, protein kinase A, ERK2, serine/threonine kinase, caspase 3 and caspase 9 also play an important role, which can be activated by Aβ [48].

### 4.3. Microglial Infiltration during Plaque Formation Leading to Neurodegeneration

In addition to extracellular plaques of Aβ and NFT due to the hyperphosphorylation of τau, microglial infiltration in response to these aggregates exacerbates the pathogenesis of AD. In addition to plaques and tangles, various morphological variants of Aβ deposits are found in the brain in AD. Extracellular and intracellular Aβ and tangles cause extreme toxicity resulting in synapse damage and increased reactive oxidative stress, which then leads to microglial infiltration around plaque areas. Microglia are resident phagocytes in the CNS and play a vital role in maintaining neuronal plasticity and synapse remodeling [51]. Microglia are activated by the accumulation of a protein that acts as a pathological trigger, migrates and initiates the innate immune response. Aβ plaques activate Toll-like receptors on microglia, which leads to microglia activation and the secretion of pro-inflammatory cytokines and chemokines [52]. In AD, microglia can bind to Aβ through cell surface receptors, including SCARA1, CD36, CD14, integrin α6 β1, CD47, and Toll-like receptors. After binding to the receptor, the microglia endocytose Aβ oligomers and NFT fibrils, which are removed by endolysosomal degradation. Microglial proteases such as neprilysin and an insulin-degrading enzyme play an important role in degradation. However, in severe cases of AD, the microglial clearance of Aβ is ineffective due to elevated local concentrations of cytokines, which suppress the expression of Aβ phagocytosis receptors and reduce Aβ clearance [48]. One of the factors underlying compromised AD clearance by microglia is a mutation in the trigger receptor expressed on myeloid cells 2 (TREM2). TREM2 mutations are associated with increased severity of AD [53].

## 5. The Model of the Pathogenesis of Alzheimer’s Disease Based on Impaired Mitochondrial Dynamics

Given the important role of mitochondrial dynamics, any changes in the precision of mechanisms can have devastating consequences for mitochondrial function, energy, and redox homeostasis. Indeed, altered mitochondrial dynamics are well documented in AD patients and model organisms with a propensity for increased mitochondrial fragmentation. Important evidence of impaired mitochondrial dynamics in AD comes from animal models of premature aging, which have impaired mitochondrial functions with the development of neurodegenerative diseases [54]. This increased fission becomes even more pronounced with a pathological increase in the levels of Aβ and pTau and their interaction with mitochondrial division regulators during disease progression [55]. Hormonal compounds can act as such mitochondrial regulators. Thus, it has been shown that women with low estrogen levels have a greater risk of developing AD [56]. In one study [57], an estrogen receptor (ER) located on the mitochondrial membrane was found to be involved in a PKA-mediated signaling pathway that leads to the inhibition of mitochondrial fission. The negative impact of Aβ on this pathway leads to increased mitochondrial fission. A study of mitochondrial morphology using brain tissue in several models of AD showed that in some cases mitochondrial fragmentation was confirmed, associated with increased levels of DRP1 and Fis1 and reduced levels of OPA1, MFN1 and MFN2 [58]. Excessive mitochondrial fission can affect energy production by affecting cristae integrity and the assembly of oxidative phosphorylation complexes, and reduced mitochondrial fusion inhibits mitochondrial repair by increasing the proportion of dysfunctional organelles. Mitochondrial fusion/fission imbalance may eventually lead to synaptic dysfunction [11]. In addition, mitochondrial fission is a precursor to apoptosis, so excessive mitochondrial fission may be directly related to neuronal death.

In AD, there is also a disruption of mitochondrial axonal transport, which is preceded by the accumulation of toxic protein aggregates and is associated with a disruption of the integrity of axons and synaptic function. The inhibition of mitochondrial mobility is closely associated with unbalanced fission/fusion regulators, elevated levels of Aβ and pTau, and oxidative stress [59]. The detrimental effect of Aβ on mitochondrial mobility was confirmed in experiments using AD models in Drosophila and mice, where the accumulation of oligomeric Aβ led to the depletion of synaptic mitochondria [59]. Additional studies on the relationship between different Aβ species and mitochondrial transport showed that toxic Aβ peptides with a higher tendency to aggregate had a greater effect on mitochondrial motility, where extracellular fibrils had the greatest influence, possibly contributing to transport abnormalities through interaction with the plasma membrane of neurons [60]. In addition to Aβ, the overexpression and/or hyperphosphorylation of Tau also impairs the distribution and localization of mitochondria in AD cells [61]. The study of the molecular mechanism of inhibition of mitochondrial transport has identified several key proteins, including Aβ and pTau, which are involved in the increased biochemical instability of mitochondria. Glycogen synthase kinase 3 (GSK3), a serine/threonine protein kinase that mediates the attachment of phosphate molecules to the amino acid residues of serine and threonine, has been shown to disrupt mitochondrial transport in AD through the phosphorylation and deactivation of mitochondrial transport motor proteins and through increased Tau phosphorylation at AT8 sites [62]. These Tau modifications not only promote the development of pTau filamentous aggregates, but also lead to increased microtubule instability, which promotes transport inhibition [63].

In neurons, most of the mitophagic degradation requires the transport of dysfunctional mitochondria within autophagic vacuoles (AVs) from distal regions to the soma. For this, autophagosomes fuse with late endosomes and form amphysomes together with dynein–snapin motors for AV transport back into the soma [44]. However, in AD, defective retrograde transport may contribute to the pathogenesis of the disease with the accumulation of amphysoma at axon terminals due to Aβ-mediated interruption of the dynein–synapsin connection. Thus, the relationship between the multiple mechanisms of mitochondrial dysfunction in AD is complex, where one (e.g., impaired axonal transport) may affect the other (e.g., mitophagy). Defective anterograde transport ensures that “young” mitochondria from the soma are unable to fuse and share their components with other mitochondria in the distal regions to promote their recovery, which leads to disruption of the energy supply within the neuron [64]. Impaired retrograde transport affects the ability to remove dysfunctional mitochondria from synaptic regions, which leads to an increase in the level of reactive oxygen species and the development of oxidative stress. Oxidative stress, in turn, promotes an increase in the expression of pro- inflammatory cytokines, which leads to impaired microglial clearance of Aβ plaques and the further progression of AD. In general, disturbances in the mitochondrial and neuronal mechanisms of transport and an increase in mitochondrial fission, combined with an increased accumulation of AVs and a reduced ability of lysosomes to fuse with autophagosomes, contribute to impaired mitochondrial quality control in AD [11]. The scheme of AD pathogenesis involving the processes of mitochondrial dynamics is shown in Figure 2.

## 6. Potential Therapeutic Strategies to Restore Mitochondrial Function in Alzheimer’s Disease

Approaches to improve mitochondrial function are divided into two groups: pharmacological and non-pharmacological interventions. Non-drug strategies are primarily aimed at shifting lifestyle towards more physically active behavior, which has a beneficial effect on improving mitochondrial function and increasing the number of mitochondria [65]. Research has shown that complex lifestyle interventions can improve cognitive function in older people at increased risk of dementia. Additional studies have confirmed the many benefits of exercise in AD patients, including improved cerebral blood flow, increased hippocampal volume, improved neurogenesis and cognitive function, reduced neuropsychiatric symptoms, and slower disease progression [66].

A promising option for the treatment of AD appears to be the use of compounds that act on mitochondrial enzymes, which induce a mild stress response, reminiscent of mechanisms associated with calorie restriction and exercise [67]. In particular, the small-molecule modulation of mitochondrial complex I activity has been found to attenuate the development of cognitive symptoms in AD when treated early in the course of the disease. Beneficial mechanisms include AMPK activation, subsequent GSK3β deactivation, decreased Aβ and pTau, the restoration of axonal transport, increased levels of brain-derived neurotrophic factor and synaptic proteins, increased bioenergetics, increased neuronal capacity to withstand oxidative stress, and the restoration of cognitive function and behavior. Widespread use of a partial inhibitor of complex I found a drug for the treatment of type II diabetes mellitus—metformin [68]. The molecular targets and mechanism of action of metformin are not fully understood. However, it inhibits energy transfer by selectively suppressing efficient binding of the redox and proton transfer domains of complex I, where subsequent AMPK activation appears to be largely associated with the main long-term clinically significant effect of increasing hepatic insulin sensitivity [68]. Clinical trials have shown that the use of antidiabetic drugs, including metformin, protects against cognitive decline in AD patients by improving executive function, learning, memory, and attention. These antidiabetic drugs positively affect mitochondrial and synaptic function, reduce neuroinflammation, and improve brain metabolism [69].

Other innovative pharmacological strategies aimed at improving mitochondrial function include antioxidant therapy, which reduces the local production of ROS in mitochondria. These compounds include coenzyme Q10, idebenone, creatine, MitoQ, MitoVitE, MitoTEMPOL, latrepirdine, triterpenoids, a set of Seto-Schiller (SS) peptides, curcumin, ginkgo biloba, and omega-3 polyunsaturated fatty acids. These mitochondria-targeting compounds have been extensively evaluated in several laboratories using a variety of in vivo and in vitro methods. The numerous benefits of these compounds include improved bioenergetics, reduced oxidative stress, and improved mitochondrial dynamics [70].

## 7. Discussion

Mitochondrial dysfunction is now recognized as a contributing factor to the early pathology of a variety of human conditions, including neurodegenerative diseases such as AD. Mitochondria are signaling organelles with a variety of functions ranging from energy production to the regulation of cellular metabolism, energy homeostasis, stress response, and cell death. The success of these complex processes critically depends on the accuracy of mitochondrial dynamics processes, which include the ability of mitochondria to change shape and arrangement in the cell, which is necessary to maintain proper functioning and quality control, especially in polarized cells such as neurons [8]. A shift in mitochondrial dynamics (extensive fission) in AD negatively affects all aspects of mitochondrial function and may be crucial for the pathogenesis of AD [11]. Therefore, strategies to alter abnormal mitochondrial dynamics may be an attractive target for therapeutic intervention in AD. Therapeutic agents aimed at reducing the expression of the mitochondrial fission proteins DRP1 and Fis1 can potentially protect neurons from energy depletion.

The search for new effective antioxidants is a promising strategy for AD therapy, as noted in the previous section. However, the main obstacle to the delivery of antioxidants/drugs is the blood–brain barrier, which provides selective permeability only for a certain set of substances. One example of such compounds is a natural plant component—saffron. Saffron may act as an antioxidant, playing a key role in protecting against CNS disease. Low cytotoxicity, commercial availability, and ability to cross the blood–brain barrier make it a suitable candidate for combating various diseases [71]. Although preclinical studies have shown promising results, the benefit of antioxidant therapy for neurodegenerative diseases in humans is still controversial. The reverse strategy, which is based on ROS-mediated hormesis, is also interesting. At low concentrations, ROS are known to be important signaling molecules useful to the cell and have the potential to protect against neurodegeneration. However, what these concentrations are is not known. In addition, it is expected that for each person these concentrations will differ, making it possible for therapeutic use only within the framework of personalized medicine [72].

## 8. Conclusions

Mitochondrial dysfunctions implicated in the pathophysiology of neuropsychiatric disorders include abnormalities in the oxidative phosphorylation cycle, increased mitochondrial DNA (mtDNA) deletions, mutations or polymorphisms, impaired calcium signaling and impaired energy metabolism, and interactions with disease-specific proteins. These processes are critically dependent on mitochondrial dynamics, the disruption of which is one of the mechanisms of pathogenesis in Alzheimer’s disease. The key aspects of pathogenesis in this case are disturbances in mitochondrial transport and the division of mitochondria, which ultimately leads to a lack of functional mitochondria in synaptic endings and an excess of dysfunctional mitochondria. Currently, several promising therapeutic strategies aimed at improving mitochondrial dynamics and function are under development. Because age is the biggest risk factor for most neurodegenerative diseases, developing or implementing health promotion strategies can delay the onset of debilitating age-related conditions, including AD.

## Figures and Tables

**Figure 1 ijms-23-06954-f001:**
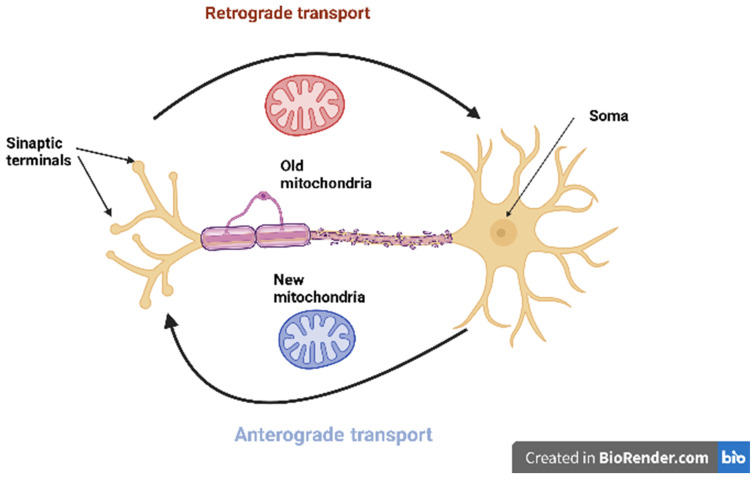
The general scheme of mitochondrial transport in neurons.

**Figure 2 ijms-23-06954-f002:**
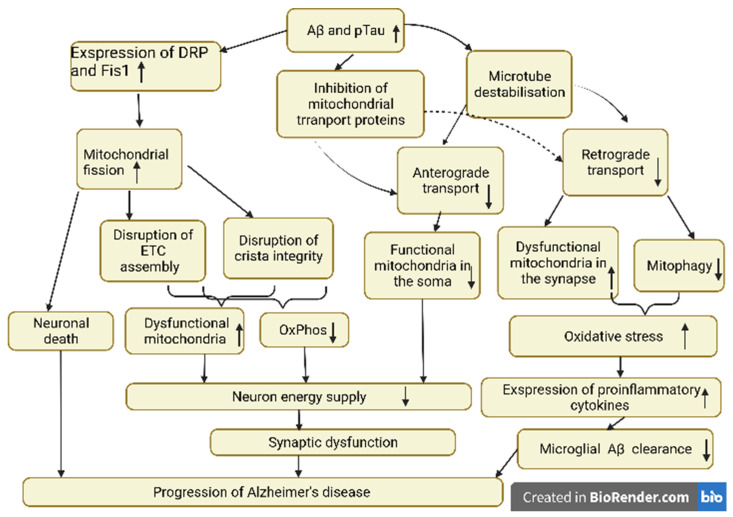
The scheme of AD pathogenesis involving the processes of mitochondrial dynamics.

## Data Availability

Not applicable.

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
