# Peer review of "Role of Impaired Mitochondrial Dynamics Processes in the Pathogenesis of Alzheimer’s Disease"

_ijms, 2022, doi:10.3390/ijms23136954_

Round 1

Reviewer 1 Report

1.      Alzheimer's disease is a progressive neurological condition associated with neuronal degeneration. Check for consistent usage of abbreviation throughout the manuscript.

2.      Line 28-34. Alzheimer's disease is a progressive neurological condition associated with neuronal degeneration, memory loss, learning disabilities, and significant changes in character and behavioral activity. Disease progression increases with age and has been 30 reported to affect 10% of people aged 65 to 75 and about 32% of people over 80. Alzheimer's disease has also been reported in young adults, sometimes as young as 20 years 32 of age (more likely to be caused by genetic mutations) [3]. By 2050, the number of people with AD worldwide is expected to reach 131 million, with the largest number of affected people expected in middle- and low-income countries. Adding few relevant publications related to AD will aid in improvement of this section. (PMID: 30898045; 32443670; 31533274).

3.      Mitochondrial dysfunction  and disruption of mitochondrial dynamics have been an attractive target for therapeutic intervention in AD in recent years. https://doi.org/10.2174/1389450121999201230204050

4.      hyperphosphorylated τau protein associated with microtubules. Is it the way tau has been represented throughout the manuscript?

5.      Figure 1 need to be improved in its presentation.

6.      Some more Figures need to be added in the manuscript. A graphical abstract needs to be embedded that depicts the summary of this work.

Author Response

Point 1: Alzheimer's disease is a progressive neurological condition associated with neuronal degeneration. Check for consistent usage of abbreviation throughout the manuscript.

Response 1: Alzheimer's disease designated as AD abbreviation in the 1-st sentence of introduction. I changed Alzheimer's disease to AD abbreviation where it needed in the text.

      Point 2:      Line 28-34. Alzheimer's disease is a progressive neurological condition associated with neuronal degeneration, memory loss, learning disabilities, and significant changes in character and behavioral activity. Disease progression increases with age and has been 30 reported to affect 10% of people aged 65 to 75 and about 32% of people over 80. Alzheimer's disease has also been reported in young adults, sometimes as young as 20 years 32 of age (more likely to be caused by genetic mutations) [3]. By 2050, the number of people with AD worldwide is expected to reach 131 million, with the largest number of affected people expected in middle- and low-income countries. Adding few relevant publications related to AD will aid in improvement of this section. (PMID: 30898045; 32443670; 31533274).

      Response 2: The publications were added.

      Point 3:   Mitochondrial dysfunction  and disruption of mitochondrial dynamics have been an attractive target for therapeutic intervention in AD in recent years. https://doi.org/10.2174/1389450121999201230204050

      Response 3: I didn't quite understand what should be changed.

.       Point 4:  hyperphosphorylated τau protein associated with microtubules. Is it the way tau has been represented throughout the manuscript?

      Response 4: This error has been corrected. “Hyperphosphorylation τau protein led to microtubule destruction.”

      Point 5:  Figure 1 need to be improved in its presentation.

      Response 5: I added new blocks in this scheme. Now it is represented as figure 2.

      Point 6: Some more Figures need to be added in the manuscript. A graphical abstract needs to be embedded that depicts the summary of this work.

      Response 6: Graphical abstract and additional figure (figure 1) were added.

Reviewer 2 Report

In part 4, the acronym for 'neurofibrillary tangles' is 'NFT'. You should correct it in several lines (214, 222, 230, 235, 236, 243, 253, 266).

In figure 1, in top panel, the letter between Abeta and pTau seems to be the russian word/letter meaning 'and'. You should change it.

Author Response

Point 1: In part 4, the acronym for 'neurofibrillary tangles' is 'NFT'. You should correct it in several lines (214, 222, 230, 235, 236, 243, 253, 266).

Response 1:  wrong acronym NFC was changed to NFT in all cases.

Point 2: In figure 1, in top panel, the letter between Abeta and pTau seems to be the russian word/letter meaning 'and'. You should change it.

Response 2:  I changed it.

Reviewer 3 Report

In this manuscript, the authors reviewed the potential role of impaired mitochondrial dynamics processes in the pathogenesis of AD. Due to the limited effectiveness of various treatments for AD, it is urgent to issue new strategies to improve the current predicament. In this regard, the authors provided some evidence and organize it to highlight that altered mitochondrial dynamics may be an important factor affecting the progression of AD, and propose therapeutic strategies to improve mitochondrial dynamics and function provide an alternative treatment approach. The ideas presented are novel and forward-looking and may be suitable for publication in this journal. But for the content, I still put forward some personal views for the authors as a reference for further revision.

1. The direct adverse effects of Aβ and tau on mitochondria dysfunction, especially the molecular mechanism, can be explained more clearly. Specifically, for example, there is evidence that intracellular Aβ can interact directly with the granulosa gland and thereby alter its function, which the authors of the narrative do not seem to mention. Since such evidence is closely related to the authors' inferences, these studies should preferably be considered for inclusion in the scope of this menuscript.

2. I don't know much about the description of "4.3 microglial infiltration...", especially the relationship between mitochondria and microglia. Is there evidence that dysfuction of microglial mitochondria leads to or aggravates AD progression?

3. The narrative on how mitochondrial dynamics are regulated can still be strengthened. In particular, it should be specified which compounds are closely related to the regulation of mitochondrial dynamics, and accordingly its future application in clinical treatment will be more prospective.

4. Mitochondrial hormesis is an important concept for the maintenance of mitochondrial function, and its relationship with AD is also very similar. The authors may also consider adding it to the discussion and explaining it.

5. Many models of premature aging in animals are induced by impaired mitochondria and are often associated with the development of AD, perhaps the authors can add this concept to their content as reinforcing evidence.

6. The content of the figure can be strengthened, especially the part related to mitochondrial dynamics.

Author Response

Point 1: The direct adverse effects of Aβ and tau on mitochondria dysfunction, especially the molecular mechanism, can be explained more clearly. Specifically, for example, there is evidence that intracellular Aβ can interact directly with the granulosa gland and thereby alter its function, which the authors of the narrative do not seem to mention. Since such evidence is closely related to the authors' inferences, these studies should preferably be considered for inclusion in the scope of this menuscript.

Response 1: I can’t find how Aβ interact with granulosa gland but I added new molecular pathway in pathogenesis AD which was connected with estrogen influence on mitochondrial dynamic regulation. (doi: 10.1016/j.brainres.2015.04.059)

Point 2: I don't know much about the description of "4.3 microglial infiltration...", especially the relationship between mitochondria and microglia. Is there evidence that dysfuction of microglial mitochondria leads to or aggravates AD progression?

Response 2: Yes, it is right. It’s good described in this article. (doi: 10.3389/fnagi.2020.00252)

Point 3: The narrative on how mitochondrial dynamics are regulated can still be strengthened. In particular, it should be specified which compounds are closely related to the regulation of mitochondrial dynamics, and accordingly its future application in clinical treatment will be more prospective.

Response 3: Please clarify: your comment refers to section 3, where I need to add cellular proteins that regulate mitochondrial dynamics or to section 6, where I need to add potential therapeutic compounds that modulate it?

Point 4: Mitochondrial hormesis is an important concept for the maintenance of mitochondrial function, and its relationship with AD is also very similar. The authors may also consider adding it to the discussion and explaining it.

Response 4: The information was added in Discussion.

Point 5: Many models of premature aging in animals are induced by impaired mitochondria and are often associated with the development of AD, perhaps the authors can add this concept to their content as reinforcing evidence.

Response 5: The information was added in section 5.

Point 6:The content of the figure can be strengthened, especially the part related to mitochondrial dynamics.

Response 6: I added the new blocks in this scheme. Now it is represented as figure 2.

Round 2

Reviewer 1 Report

The manuscript can now be accepted for publication.